# Building a Chinese pan-genome of 486 individuals

Qiuhui Li[1,3], Shilin Tian [2,3], Bin Yan[1,3], Chi Man Liu[1,3], Tak-Wah Lam[1✉], Ruiqiang Li[2✉] & Ruibang Luo [1✉]

Pan-genome sequence analysis of human population ancestry is critical for expanding and better defining human genome sequence diversity. However, the amount of genetic variation still missing from current human reference sequences is still unknown. Here, we used 486 deep-sequenced Han Chinese genomes to identify 276 Mbp of DNA sequences that, to our knowledge, are absent in the current human reference. We classified these sequences into individual-specific and common sequences, and propose that the common sequence size is uncapped with a growing population. The 46.646 Mbp common sequences obtained from the 486 individuals improved the accuracy of variant calling and mapping rate when added to the reference genome. We also analyzed the genomic positions of these common sequences and found that they came from genomic regions characterized by high mutation rate and low pathogenicity. Our study authenticates the Chinese pan-genome as representative of DNA sequences specific to the Han Chinese population missing from the GRCh38 reference genome and establishes the newly defined common sequences as candidates to supplement the current human reference.

[1] Department of Computer Science, The University of Hong Kong, Hong Kong, China. [2] Novogene Bioinformatics Institute, Beijing, China. [3]These authors contributed equally: Qiuhui Li, Shilin Tian, Bin Yan, Chi Man Liu. ✉email: twlam@cs.hku.hk; lirq@novogene.com; rbluo@cs.hku.hk

In genome-based studies, the creation of highly enriched reference sequences is fundamental for dissecting the associations of genetic variants with human diseases. To capture these variations as much as possible, the genomics field has been trying to build and perfect human reference genomes. In the majority of studies, for different species, the approach is still to rely on individual sequence assessments as a reference for each species. However, the use of a single reference genome often misses important variations or fails to detect sequence differences[1]. The current reference assembly for humans, GRCh38, is being improved continually by researchers around the globe, but it still under-represents the human population. Currently, it is constructed mostly from individual sequences of Caucasian and African ancestries[2,3], which effectively limits genomic analysis of other distinct population ancestries. As a consequence, some population-specific genomic variants cannot be detected when aligned to the current reference. This lack of vital information on genetic diversity makes it almost impossible to uncover some genetic links with disease.

To increase diversity of the current reference genomes in human, several personal genomes have been assembled from subpopulations, including Korean, Caucasian, and Mongolian genomes[4–6]. Others have attempted to assemble reference genomes across diverse populations, such as the Simons Genome Diversity Project[7] and the TOPMed Project[8]. Still other projects have used the genomes of homogenous populations to discover novel sequence insertions. The Genome of the Netherlands Project released 7718 novel segments[9], while 3791 novel non-repetitive sequences were called from 15,219 Icelanders[10]. While the identification of sequence deletions or insertions may be beneficial for exploring evolutionary directions[7], for understanding the relevance of abnormally expressed genes and novel sequences[11] and for discovering diseases specific to a particular population ancestries[10], these data have not been integrated into a new, more representative sequence reference for the human genome.

In order to acquire a more complete picture of genetic variations, "pan-genome" coverage has been proposed of both the core and variably distributed genomes of a species[12]. Pan-genome is a collection of all DNA sequences that occur in a species. Ideally, pan-genomic analysis better captures unexplored or missed variants to improve the decoding of the genetic basis of human diseases[1]. Advances in sequencing and assembly technologies, as well as the decline in sequencing costs, facilitate pan-genome studies across a large range of species, from bacteria to plants to humans[13–18]. Considering the computational complexity of assembling deep sequenced human genomes *de novo* and combining them, a feasible strategy for improved references is to organize population-specific pan-genomes and to extend genetic diversity of the existing pan-genomes. In this regard, two human pan-genomes have been published. The first whole-genome sequencing and assembly project for human pan-genome was reported on in 2010. This project identified approximately 5 Mbp of novel sequences absent in the reference genome for each of the assemblies and predicted that the total size of the novel sequences reached 19–40 Mbp[19]. Later, a project to create an African pan-genome (APG) derived from 910 African genomes detected 296 Mbp of novel DNA sequences, considerably more than the 2010 project[20]. A more recent report named HUPAN, a human pan-genome construction and analysis system that had built the first pan-genome of Chinese, identified 29.5 Mbp of Chinese-specific novel sequences derived from 275 individuals[21]. These studies show that the sizes of the pan-genomes in humans are variable and therefore ill-defined and the studies do not indicate whether pan-genomes grow indefinitely as more individual inputs are sequenced. In other words, it remains unclear

whether the size of pan-genomes will reach a maximum or continue to grow with increasing number of individuals. More importantly, biological interpretation of the predicted sequences needs to be implemented for defining the pan-genomes.

In this study, we used deep sequencing data of 486 Han Chinese individuals to build a Chinese pan-genome (CPG), consisting of 276 Mbp DNA sequences that are absent from the current human reference. The novel sequences could be classified into two components: individual-specific sequences with an average of 0.472 Mbp and common sequences (novel sequences shared by at least 2 individuals) totaling 46.646 Mbp. Based on this two-component model, we showed that the total size of common sequences could increase linearly with the number of individuals, hence constructing pan-genomes for huge populations would be impractical. Nevertheless, we also showed that by defining "common sequences" as sequences whose occurrence within a population is above a fixed percentage, their total size would remain constant regardless of individuals. Finally, for applications, the distinct characteristics of the CPG common sequences improved variant calling and sequence alignments, and could be used to identify special regions with high mutation rates and low pathogenicity. Thus, the common sequences defined in CPG could be used as new "decoys" and effectively increase the diversity of current human reference genome.

## Results

We selected whole-genome sequencing datasets from a pool of short-read sequencing datasets of normal Han Chinese individuals from the Novo-Zhonghua Genome Database. It is a commercial reference database of Chinese by Novogene Co., Ltd., Beijing, China, and has been used for multiple studies[22–27]. Other than ethnicity as "Han Chinese", all individuals were de-identified and personal details were unavailable. We used verifyBamID[28] to detect sample contamination and removed any individuals with an "estimate of contamination" greater than 0.03. We then analyzed the remaining 1000 genome project individuals using Principal Component Analysis and excluded any that significantly deviated from the CHB (Han Chinese in Beijing) group. In total, 486 individuals were employed in this study, with an average sequencing depth of 53.6-fold (min 27.1-fold, max 121.3-fold).

We used ADMIXTURE (version 1.3.0)[29] to analyze the population components of the 486 individuals (Fig. 1a, Supplementary Method). At $k = 2$, the result shows a highly consistent composition of the 486 individuals, while at higher $k$, subtle diversities emerged. This indicates that the 486 individuals remained within the scope of a single ethnic (Han Chinese) well and were representative of the diversities of the population.

**A workflow for defining a novel pan-genome**. To construct a Chinese pan-genome (CPG), we have established an analysis system that carried out four major tasks (Fig. 2). Firstly, we aligned the sequencing reads of 486 Han Chinese to the GRCh38.p13 reference genome individually and gained reads that could not be mapped to the reference genome. The unaligned reads were assembled into contigs (continuous sequences). Any contigs identified as contaminants or mapped to GRCh38 were eliminated. We then calculated the novel sequence size of every individual after removing the contaminated DNA sequences. In the size distribution of the novel sequences, most of the individuals are located within a range of 2.5–3.5 Mbp (Fig. 1b). Based on the alignment positions of contigs' reads and mates to GRCh38, we classified these contigs into two types: placed and unplaced ones. The exact insertion breakpoints of the placed contigs were determined and the placed contigs were then separated into three parts: right-end-placed (REP), left-end-placed

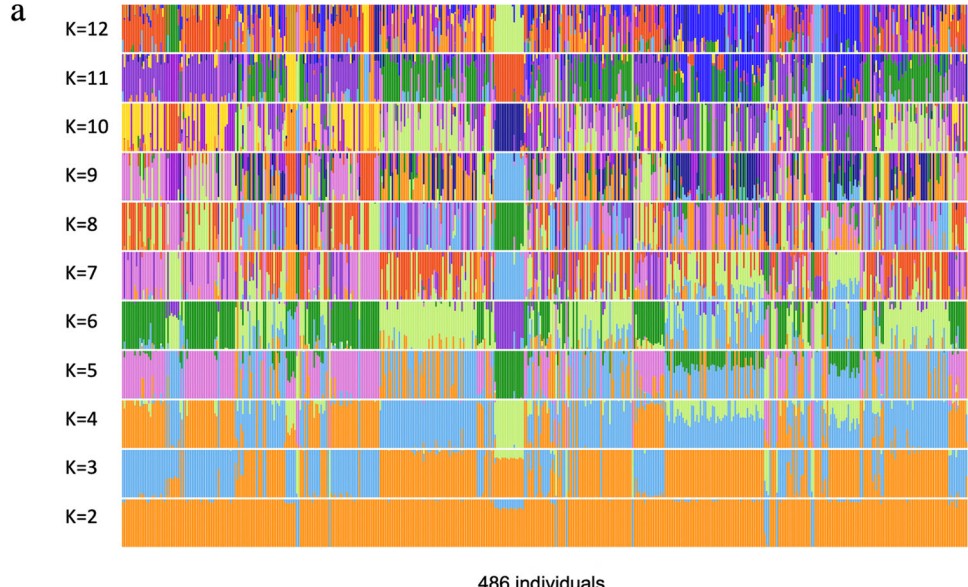

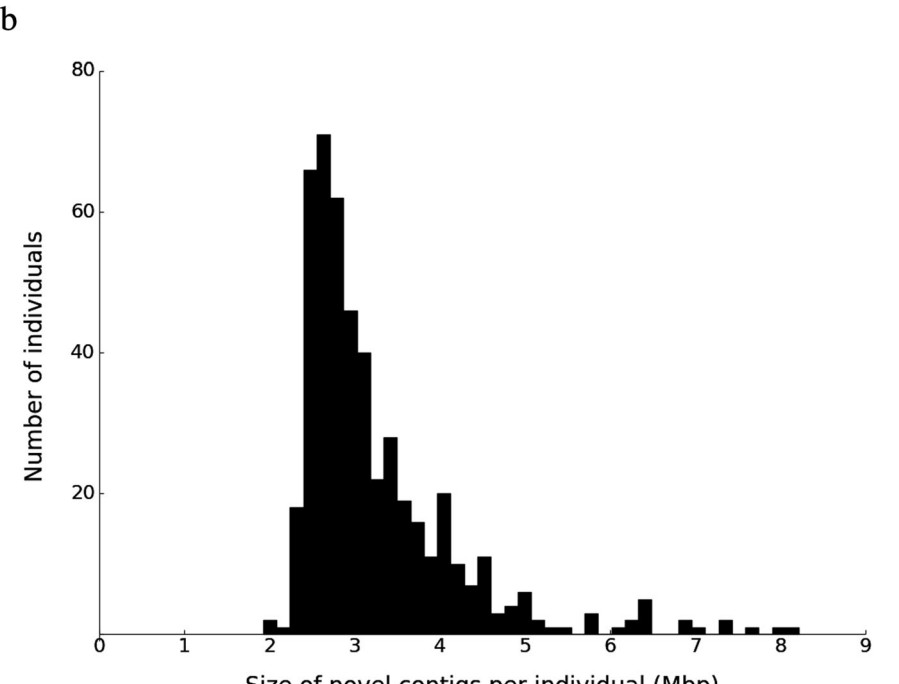

**Fig. 1 Composition and novel sequence size of the 486 Chinese individuals. a** An overview of the population components of the 486 individuals. We carried out the unsupervised ADMIXTURE analysis and showed the highest-likelihood replicate for different numbers of ancestral populations K. Each vertical column represents the proportion of ancestries in K constructed ancestral populations for each individual. **b** Distribution of novel sequence length in 486 individuals. The smallest and largest sizes of novel sequences in an individual are 1.92 Mbp and 8.2 Mbp, respectively. For the source of the novel sequences, please refer to the "Data availability" section.

(LEP), and both-end-placed (BEP). Secondly, we compared the placed contigs against one another and clustered together the similar contigs that placed close to each other. Unplaced contigs aligned closely to the placed ones with high identities were also included in the placed clusters (all the placed sequences see Supplementary Data 1). The remaining unplaced contigs were clustered by the cd-hit-est method[30] (all the unplaced sequences see Supplementary Data 2). More definitions and details of placed and unplaced contigs are in Methods. We identified a total of 612,914 novel sequences, including 276,116,245 bp not present in the GRCh38 reference genome. Thirdly, we classified the identified novel sequences (~276 Mbp) into two components:

individual-specific and common sequences. We then estimated the growth of the two types of sequences, especially the common component. Finally, through characterization, we identified the novel sequences capable of representing genomic regions of Chinese missed from the current human reference, and also applied the common sequences to improve variant calling and sequence mapping. After completion of these four steps, we concluded that the novel sequences could be used to define the Chinese pan-genome, CPG.

In order to ensure that only non-contaminants were included in our analyses, we aligned all of the novel sequences to the NCBI Nucleotide database by BLASTN[31]. The result showed only 0.3%

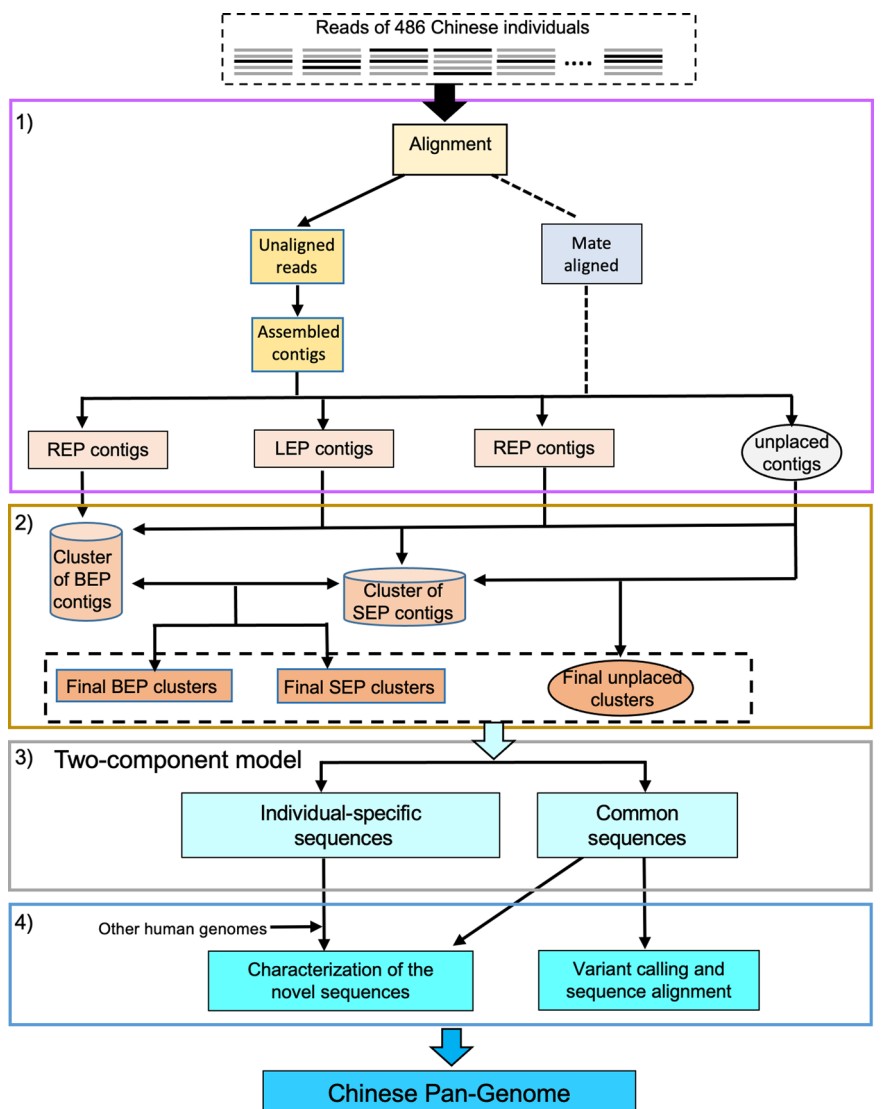

**Fig. 2 Workflow for construction of a Chinese pan-genome (CPG).** There are four procedures for construction of CPG, (1) generation of four types of contigs, right-end-placed (REP), left-end-placed (LEP), both-end-placed (BEP), and unplaced contigs; (2) identification of the novel representative sequences; (3) modeling of two-component common and individual-specific sequences; (4) further analysis of Chinese Pan-Genome. The first two procedures were in accordance with the APG method.

of CPG aligned to nonhuman sequences (Supplementary Note 1, Supplementary Data 3). Common sequences, defined as sequences shared by at least 2 out of 486 individuals, had a total size of 46.646 Mbp, making up 16.9% of all the novel sequences in CPG (Table 1); the remaining 83.1% were individual-specific sequences. We found that less than 3.2% of novel sequences could be aligned to GRCh38 (with 80% identity and 50% coverage), confirming the novelty of CPG (Table 1). Among all novel sequences, the breakpoints of 13,544 sequences were resolved. We then investigated how many individuals of the 486 Chinese shared the novel sequences (Table 1). We determined that the breakpoint-resolved sequences were shared among more individuals than the unplaced sequences.

**Chinese pan-genome size**. Here we propose a two-component model for CPG. In this model, every novel sequence is classified into one of the two classes: individual-specific or common. For our 486-person population, a novel sequence is regarded as a "common sequence" if it appears in at least 2 individuals;

otherwise it is an "individual-specific sequence". We chose the number 2 as the occurrence threshold due to the small population size and we will discuss later how this threshold affects the pan-genome size. To study the effect of population size on these two classes of novel sequences, we plotted their growth curves with respect to population size (Fig. 3a). From the plot, we see that the individual-specific component increased linearly at a rate of 0.472 Mbp per individual. In contrast, the common component grows quickly at the beginning and exhibits gradual slowdown as the number of individuals increases, suggesting that an almost-complete (say, 90%) set of common sequences can be obtained by even fewer individuals (in our case, 279). From Fig. 3a alone, it is unclear whether the common sequences will grow indefinitely, or reach a maximum at some point when more individuals are added to the population. Intuitively, using an occurrence threshold larger than 2 can reduce the total size of common sequences. To illustrate the effect of occurrence threshold on common sequences, we generated common sequences at different thresholds from 5 to 200. The corresponding sizes of these shared sequences were calculated for increasing numbers of individuals

**Table 1 Novel sequences in the Chinese pan-genome.**

| | Number of sequences | Total size (bp) | Bases with not alignment to GRCh38[a] | Longest sequence (bp) | Average number of sequences per individual | Average number of individuals per sequence |
|---|---|---|---|---|---|---|
| Both-end-placed | 527 | 338,763 | 337,660 | 4420 | 67.75 | 62.48 |
| Single-end-placed | 13,017 | 6,885,137 | 6,511,621 | 20,514 | 150.38 | 5.61 |
| Unplaced | 599,370 | 268,892,345 | 260,478,456 | 239,439 | 2015.16 | 1.63 |
| Total | 612,914 | 276,116,245 | 267,327,737 | 239,439 | 2233.30 | 1.77 |
| Common | 45,836 | 46,645,513 | 45,453,460 | 128,712 | 1066.47 | 11.31 |
| Individual | 567,078 | 229,470,732 | 229,470,732 | 239,439 | 1166.83 | 1.0 |

a<80% identity or <50% coverage of the whole sequence.

(see "Method"). These calculated shared sequence sizes are plotted in Fig. 3b which displays a set of increase curves from linear to slow with increase in population size or occurrence thresholds. This means that for a constant size of population, there are fewer common sequences as the thresholds become larger; but for a constant occurrence threshold, there are more common sequences as the population increases. However, it is still not obvious from Fig. 3b whether the total size of common sequences can increase indefinitely when a larger occurrence threshold is used. In fact, using a simplified tree model for population growth, we showed that the total size of common sequences could increase linearly with respect to population size (Supplementary Note 2) whenever a constant occurrence threshold is used. This led us to the idea of using a ratio instead of a constant value as the threshold.

Figure 3b suggests that the size of common sequences is likely determined by the ratio of occurrence threshold to population size, here referred to as "occurrence frequency" (OF). For example, when threshold = 100 and population = 400, there are 1.163 Mbp of common sequences; when threshold = 50 and population = 200, there are 1.156 Mbp; and when threshold = 25 and population = 100, there are 1.149 Mbp. In other words, when OF is 0.25 or 25%, the common sequences have roughly total size of 1.15 Mbp. A short explanation for this phenomenon can be found in Supplementary Note 2. The growth of the common sequences is also analyzed in Supplementary Note 2. It is worth mentioning that the estimated size of common sequences corresponding to the same OF was different for CPG and APG (Fig. 3c), due to reasons such as different ethnicities and data processing methods. Such discrepancies must be taken into account when using OF to estimate the size of common sequences for larger populations.

**Characterization of the novel sequences.** To analyze the functional difference between the two components, we computed pathogenicity scores using the C-scores reported by Combined Annotation-Dependent Depletion (CADD). CADD[32] is a widely used database of pathogenicity (i.e., deleteriousness) of mutations at each genomic position. We found that the mean pathogenicity scores of the insertion regions of the common and individual-specific sequences were 4.295 and 5.126, respectively (Fig. 4a). A higher C-score means more deleterious outcomes if having a mutation at an insertion point. The pathogenicity score of the common sequences' insertion regions was lower than 4.83, the average of whole GRCh38. Under the assumption of neutral drift, our result suggests that the common sequences inserted into the low-pathogenic regions tend to be benign-likely.

Our analyses show that the novel sequences were not randomly inserted into the reference genome. Instead, most of these sequences were inserted into regions with a high mutation rate (Fig. 4b, Supplementary Note 3). Although the mutation rate of our insertion points was the highest, the corresponding pathogenicity at the insertion points was the lowest (Fig. 4c). We further observed enrichment of novel sequences in simple repeat and satellite regions (Supplementary Table 1), consistent with the previous studies. Simple sequence repeats have a key effect in generating variants underlying adaptive evolution[33,34]. The novel sequences were also enriched with the regulatory regions, such as CTCF binding sites and promoters (Supplementary Table 2, Supplementary Note 4).

To investigate the functional relevance of the novel sequence, we used VEP (release 98)[35] to annotate the insertion points. We determined that there were 124 placed sequences falling within the coding sequence regions intersected with 80 protein-coding genes (Fig. 5, Supplementary Note 5). This result implies a

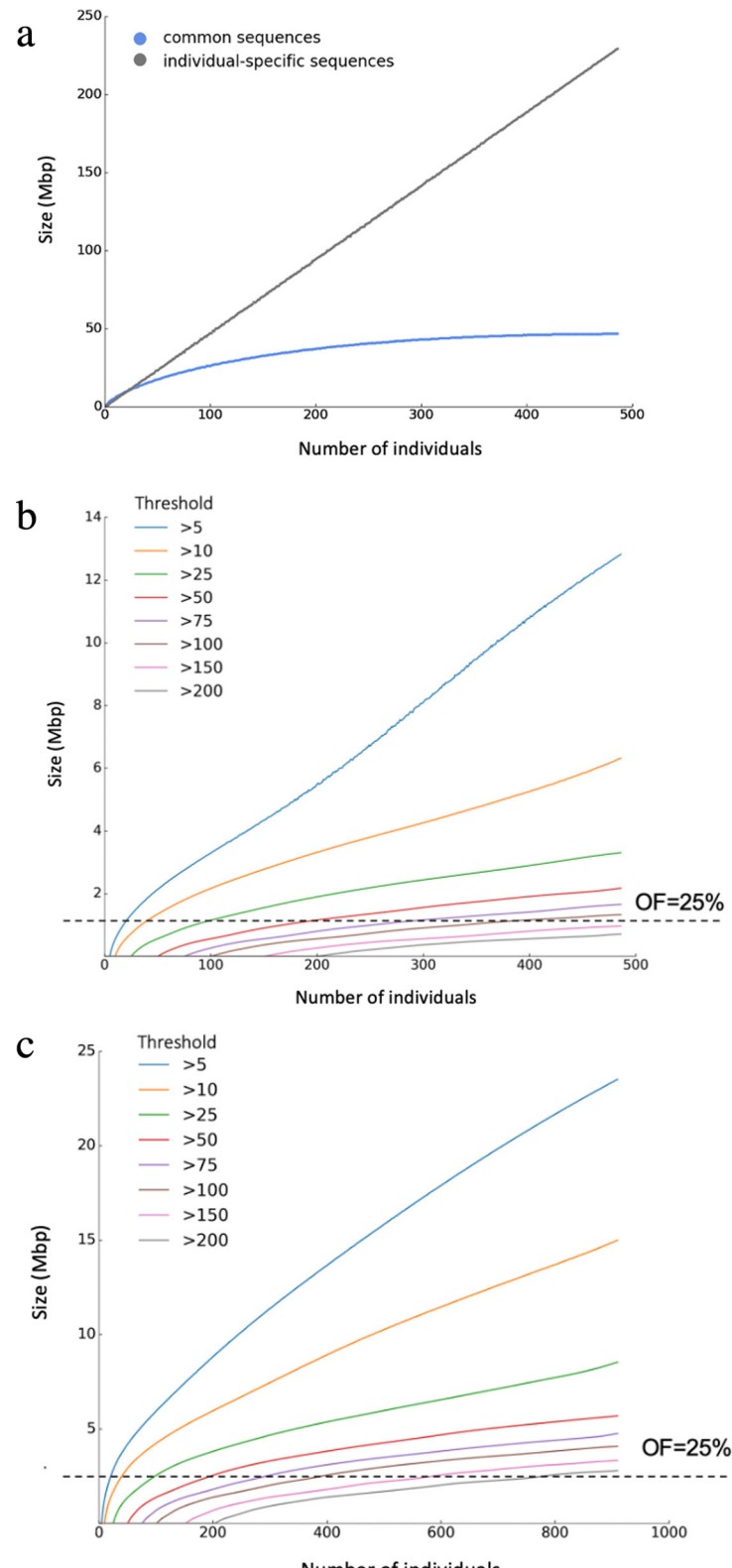

possibility of the placed sequences in affecting expression of these genes and calls for a large-scale gene expression profiling in Chinese for verification.

To seek the origin of the sequences, we aligned CPG to the chimpanzee genome (GCF_002880755.1) by BWA (version 0.7.12). We considered a novel sequence to be of chimpanzee origin if it aligned to the chimpanzee genome with over 90%

identity and 80% coverage. 0.96% of the novel sequences passed the thresholds, suggesting that these sequences are ancestral to humans. Of the individual-specific sequences, 0.47% (compared to 7.05% of the common sequences) was aligned to chimpanzee.

To look for relationships with other human populations, we aligned the novel CPG sequences with those available from three human whole genomes, HX1 (Chinese), KOREF1.0 (Korean), and

**Fig. 3 Growth of Pan-genome sequences. a** The sizes of individual-specific and common sequences grew with increase in individual number under a fixed 486 people of the CPG population. Every blue dot represents the average size of the common sequences (shared by 2 individuals or occurrence threshold = 2) in a certain number of individuals after randomly sampling 1000 times (see "Methods"). The corresponding size of the individual-specific sequences (black dots) were obtained by total Pan-genome size minus the common sequence size. **b** Growth of common sequence sizes with increase in individual number of the CPG population. **c** Growth of common sequence sizes with increase in individual number of the APG population. In **b**, **c** the common sequences shared by different thresholds from 5 to 200 individuals. Each point was obtained by randomly sampling 1000 times from the CPG or APG population and then counting the total size of novel sequences (see "Methods"). OF, occurrence frequency = a ratio of the threshold to population size.

NA18507 (African)[4,11,36]. Based on a minimum requirement of 90% identity, the percentages of alignments to Chinese (23.7%) and Korean (24.5%) were much higher than to African (16.3%) (Fig. 6a). This provides evidence that our novel sequences are of Chinese or of East Asian ancestry. We further compared our CPG to two existing pan-genomes, APG[20] and the Han Chinese pan-genome (HCPG)[21]. We aligned CPG to APG and HCPG using thresholds of 90% identity and 50% coverage and vice versa. CPG aligned to both APG and HCPG with a low mapping rate of less than 10% (4.89% (13.51 Mbp in size) and 5.04% (13.93 Mbp in size)). When aligning the common sequences in CPG to HCPG, the mapping rate increased to 14.5%. Relatively, alignment percentage of HCPG to CPG was 8.4%, higher than 1.8% of the APG to CPG (Fig. 6b, c). This result indicates the specificity of our pan-genome among Chinese and the genetic differences between the Chinese and African genomes.

We predicted 53 protein-coding genes from our novel sequences (Supplementary Note 6). Analysis of the novel sequences at protein level produced 24,300 hits to 6,138 RefSeq human proteins satisfying ≥70% identity and $<1 \times 10^{-10}$ e-value (Supplementary Fig. 1). The result shows that protein function of over 90% of the novel sequences is unknown. We found that 266 of the placed sequences were mapped to the Pfam 32.0 dataset with <0.001 e-value (Supplementary Fig. 2). Similarly, most of protein families that the placed sequences annotated are still unknown. Furthermore, we used RepeatMasker to identify the repeat sequences. Among the novel sequences, the percentages of short interspersed nuclear elements, long interspersed nuclear elements, and satellites were significantly higher than those of other repeat types (Supplementary Fig. 3, Supplementary Note 5). We used dna-brnn[37] to identify centromeric sequences in CPG. dna-brnn was used on APG and identified 85.1% of its sequences to be centromeric (82.4% Human Satellites 2 and 3 (hsat2,3); 2.7% alphoid satellite DNA (alphoid)). However, it's not the case in CPG. dna-brnn shows that only 0.5% (0.3% hsat2,3; 0.2% alphoid) and 0.7% (0.4% hsat2,3; 0.3% alphoid) of the bases in common and individual sequences, respectively, were centromeric (Supplementary Table 3). We detected a higher GC content of the novel sequences than that of GRCh38 (40.9%): The placed sequences had 47.7% GC content slightly lower than the unplaced ones (49.8%), which is in agreement with the previous studies (Supplementary Fig. 4)[7,19,21].

**The applications of the common sequences**. We sought to investigate whether the common sequences can enhance the performance of sequence alignment. To do this, we aligned the unmapped reads of each individual to the common sequences (Supplementary Note 7). It is noteworthy that over 90% of the unaligned reads from the 486 individuals would be mapped to these common sequences (Fig. 7a), which is only equivalent to 1.5% of the size of GRCh38. To further verify our findings, we downloaded the assemblies of 90 Han Chinese (45 CHB and 45 CHS) genomes from a previous study[38]. We derived the novel sequences from the 90 genomes and aligned them to our common sequences (Supplementary Fig. 5, Supplementary Notes 8, 9). The

alignment shows that 88 individuals can map >80% of their novel sequences to our common sequences with ≥80% identity (Fig. 7b). This result highlights the representativeness and the potential of our common sequences to increase mapping rate in Han Chinese significantly.

An important application of novel sequences is to improve the quality of variant calling. We combined the common sequences with the full GRCh38 (including all unplaced contigs, alternative contigs, decoy sequences and HLA subtype sequences) as a new reference (Supplementary Note 7). We aligned 60-fold sequencing reads of individuals HG001 (Caucasian), HG002 (Ashkenazim), and HG005 (Han Chinese)[39,40] to the new reference using BWA-mem[41] and called small variants using GATK HaplotypeCaller[42] with its best practice guidelines. We evaluated the variants and obtained the number of true positive, false positive, and false negative using RTG vcfeval against GIAB v3.3.2 truth variants. A significant decline in false-positive variants with negligible changes in true-positive and false-negative variants was observed. The number of false-positive variants reduced by 4.78%, 3.32%, and 4.73%, in HG001, HG002, and HG005, respectively (Table 2). These findings indicated that the new reference improved the performance of variant calling not only for Han Chinese but also for other ancestries. Since the common sequences we added to the reference are from the Han Chinese population, we had expected a more significant improvement in HG005 than HG001 and HG002. This could be explained by a relatively lower number of truth variants in HG005 recorded in GIAB[43], which is confirmed by our observation that the number of true positive variants in HG005 is only about 3.25 million, but HG001 has 3.48 million and HG002 3.44 million. We further increased the sequencing depth by five times from 60-fold to 300-fold for HG005 and notably, the improvement persisted (false-positive variants reduced by 5.19%), indicating that the improvements were indeed from adding the common sequences instead of insufficient sequencing depth.

## Discussion

A major goal for pan-genome research in genetics is to develop reference sequences that are able to capture highly enriched information of genetic variations and eventually to dissect complex associations between the variations and human diseases. Because of human subpopulation diversity and complexity of human genomes, finding a widely accepted standard for building pan-genomes remains challenging[1,44]. In this study, we assembled the genomic reads from 486 Han Chinese individuals and identified a set of 276 Mbp DNA sequences. We classified the novel sequences into two components, the common and individual-specific sequences and proposed an uncapped growth model of the common sequence size. We have shown the specific characteristics and application of the common sequences, in particular, the improvement in variant calling and efficient alignment and insertion into special genomic regions. These results provide evidence that the newly identified 46.646 Mbp common sequences could represent shared genomic information in the Han

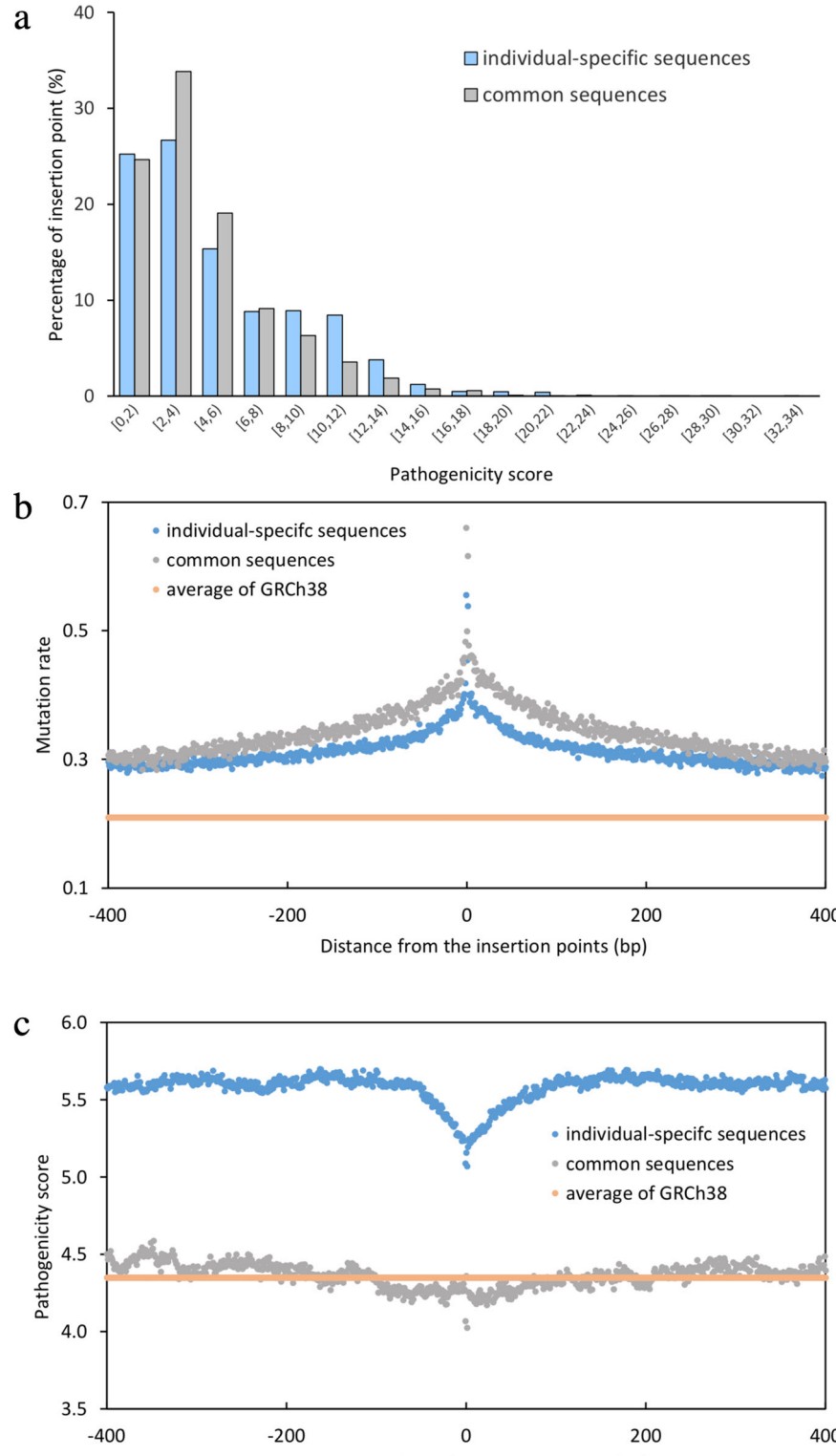

**Fig. 4 Characterizations of the common sequences and insertion points. a** The percentages of the insertion points of the common and individual-specific sequences in each binning of pathogenicity scores. **b** The mutation rate in the 500 bp flanking region of insertion points. **c** The pathogenicity score in the 500 bp flanking region of insertion points. The average pathogenicity score was calculated using GRCh38 data. The average pathogenicity scores of the insertion regions of common and individual-specific sequences were calculated (see "Methods").

Chinese population missing from the current human reference genome.

One challenge in human pan-genome research is to determine whether there exists an upper bound limit to the size of a pan-genome. This leads to a further question of how many individuals are needed for finding population-specific common sequences. Previous analyses reported that some species like human pathogens and bacteria have an open pan-genome, presuming an infinite model of the pan-genomes in these species[45]. Our previous report suggested that there is an upper bound limit to the

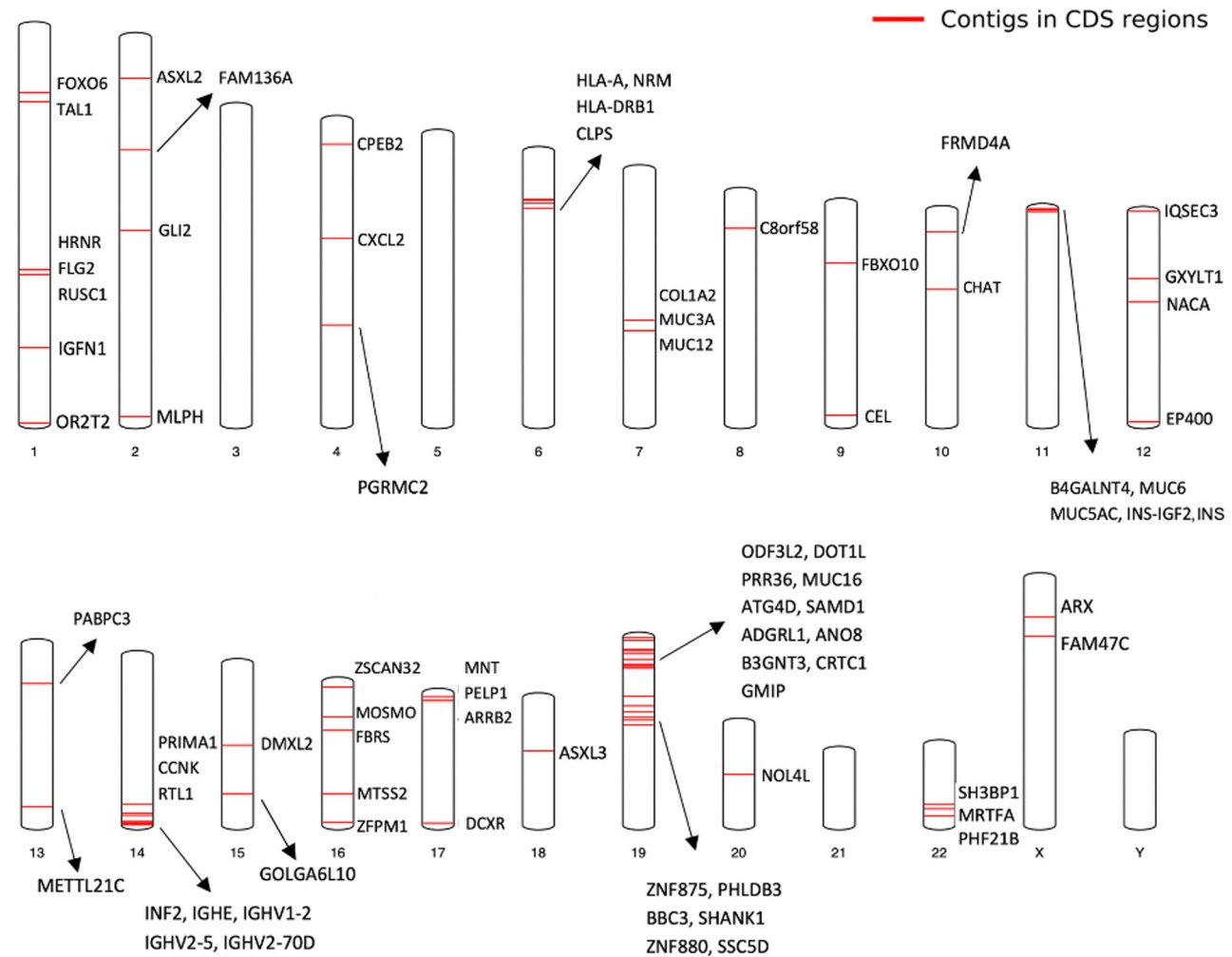

**Fig. 5 Chromosome locations of the novel CPG sequences inserted in CDS regions.** In the genome map, the red lines represent the novel CPG sequences located in coding sequence (CDS) regions and the corresponding gene symbols are listed. Detailed annotation information was reported in Supplementary Data 6.

human pan-genome size[19]. However, a recent study observed an almost linear growth of the APG size when more sequenced individuals were added[20]. Our proposed two-component model for pan-genomes suggests that this linear growth is likely due to the individual-specific sequences. Under the two-component model, it is no longer relevant to discuss an upper limit of the pan-genome size unless we limit our scope to only common sequences. Using the HUPAN (a Han Chinese pan-genome built by the HUman Pan-genome ANalysis system) analysis pipeline[21], the size of novel sequences in 50 individuals was 22 Mbp and was further increased only by 4 Mbp and 3 Mbp, respectively with the individual numbers expanded to 100 and 150. Although HUPAN's novel sequences were not explicitly classified into individual-specific and common, the slowdown in increase might suggest that the total size of common sequences was converging and eventually bounded. Similar slowdown patterns were also observed in the common sequences of CPG, but it was difficult to draw a definite conclusion since the sample size (486) was too small compared to the whole Chinese population. We noticed that the occurrence threshold was an important factor for estimating the total size of common sequences and proposed to use a proportionate occurrence frequency, OF, when defining common sequences. Our analysis showed that the total size of common sequences was bounded if OF was being used and became unbounded if a constant occurrence threshold was used instead,

which was not in agreement with our previous opinion[21]. However, it is worth mentioning that the common sequence sizes estimated from CPG and APG are not similar under the same OF. The main reason is possibly related to composition of populations, sequencing technologies, and analysis methods, etc. In APG, more individuals are needed to find the common sequences in the African population due to a higher heterogeneity. The slower growth observed in HUPAN might be due to the genome assembler used in their study. HUPAN has used 185 deep sequencing and 90 assembled Han Chinese genomes. While the 185 deep sequencing genomes were PCR free and assembled using SGA[46], the 90 assembled Han Chinese were sequenced much earlier in 2013 with a PCR-based protocol and assembled using SOAPdenovo2[38]. For PCR-based sequencing data—the DNA sequences that contain either excessively high or excessively low GC may be partially missed from the result. SOAPdenovo2[47] was developed by the corresponding author of this study. SOAPdenovo2 is an assembler that maximizes sequence contiguity in trade of sequence completeness, especially for those repetitive sequences that are more commonly observed in novel sequences. In the present study, we used only PCR free genome sequences and assembler MEGAHIT, which was designed to maximize sequence completeness. We expected the use of PCR free data and MEGAHIT will largely maximize the sequence completeness of this study, and we indeed observed a larger size of novel

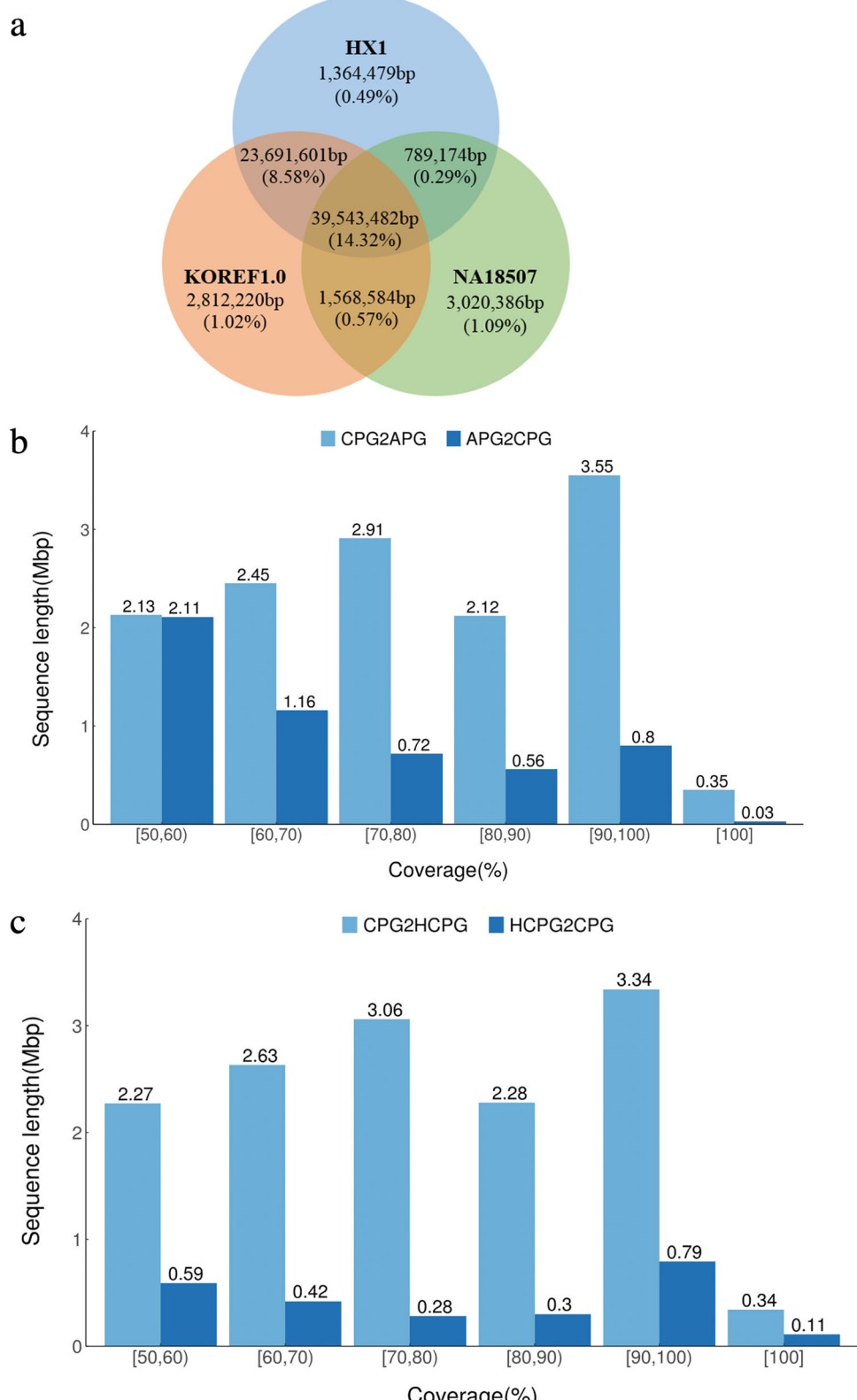

**Fig. 6 Alignment of CPG to other human genomes. a** Venn diagram of alignment of CPG to three human genomes, HX1 (Chinese), KOREF1.0 (Korean), and NA18507 (African). The alignments were based on at least 90% identity. **b** The length distribution of novel sequences from APG and CPG that can be aligned with each other with ≥90% identity. CPG2APG: CPG aligned to APG. APG2CPG: APG aligned to CPG. **c** The length distribution of novel sequences from CPG and HCPG that can be aligned with each other with ≥90% identity. CPG2HCPG: CPG aligned to HCPG. HCPG2CPG: HCPG aligned to CPG.

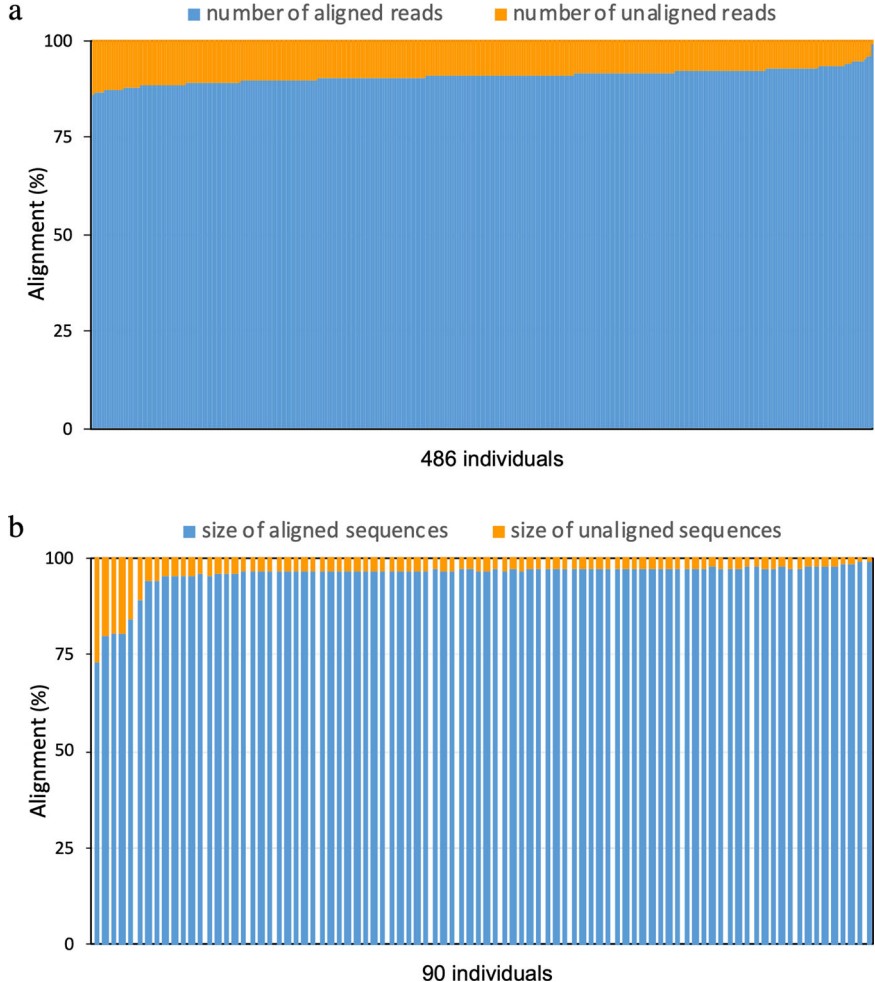

**Fig. 7 Percentage of alignment to the common sequences. a** Alignment of the unaligned reads from 486 individuals to the common sequences. **b** Alignment of the novel sequences from 90 Han Chinese to the common sequences. Each column represents an individual. The 90 genomes were extracted from WGS of Han Chinese genomes project[38].

| Table 2 Variant calling performance with different reference genomes. | | | | |
|---|---|---|---|---|
| **Dataset** | **Reference Genome** | **#$_{True-pos}$** | **#$_{False-pos}$** | **#$_{False-neg}$** |
| HG001 (60X) | GRCh38 | 3,478,662 | 12,819 | 63,679 |
| | GRCh38 + CPG | 3,478,639 | 12,206 (4.78% lower) | 63,702 |
| HG002 (60X) | GRCh38 | 3,443,252 | 16,575 | 61,707 |
| | GRCh38 + CPG | 3,443,250 | 16,025 (3.32% lower) | 61,709 |
| HG005 (60X) | GRCh38 | 3,249,098 | 9895 | 59,116 |
| | GRCh38 + CPG | 3,249,085 | 9,427 (4.73% lower) | 59,129 |
| HG005 (300X) | GRCh38 | 3,252,032 | 6900 | 56,182 |
| | GRCh38 + CPG | 3,252,026 | 6542 (5.19% lower) | 56,189 |
| "CPG" represents the 46.646 Mbp common sequences. | | | | |

sequences. But we also realized that not all these additional novel sequences are correct, and they need to be further validated through long-read assemblies or even wet-lab experiments in the future.

In general, current human genome analysis starts by aligning unknown sequences to the current references and then discarding the unmapped data. This traditional method fails to detect some disease-related mutations and other important genetic variant sites that are harbored on the unaligned sequences. Previously, Simons Genome Diversity Project identified 5.8 Mbp of novel

sequences and reported that these sequences can improve read mapping[7]. However, this project did not provide the detailed description for the result. Our common sequences have significantly elevated the alignment rate in both our datasets and public human genome information. This finding supports the possibility of the common sequences of CPG as representative of Han Chinese genomes missing from the current reference. Furthermore, a previous study used multiple reference genomes to replace a single reference genome for improving variation calling on human genome[48]. However, that study was

inconclusive due to testing only low-depth (10×) sequencing reads, which are prone to excessive amounts of false-positive and false-negative variants. We used 60-fold and 300-fold depth sequencing reads from three populations to call variants. The improvements were consistently observed in all three populations and when increasing from normal-depth to ultra-high depth.

The insertion points of the common sequences held the highest mutation rates and the lowest pathogenicity scores, which were even lower than the average pathogenicity of GRCh38. A low pathogenicity indicates that the mutation at an insertion point is relatively benign. Mutation is a raw material of evolution and the change in DNA sequences are likely beneficial to its organism[49]. Under natural selection, benign mutations are tolerated and inherited by the next generations. Thus these benign-likely sequences became common ones. The characteristics of the CPG common sequences may be explained by their relationships with population-specific evolution. Evolution is the process by which a species must overcome new conditions and create a new set of unique metabolic reactions to a particular environmental stress[50]. As humans dispersed out of Africa, the subpopulations have experienced selective pressures affecting genome integrity. Under this situation, a high mutation rate is favored to produce novel genotypes at loci which are responsible for adaptation to new environments[51]. The adaptive response could accelerate the evolutionary rate of the lineage and the functional evolution of specific stress-response proteins[52,53]. We presume that the high mutation rate of the CPG common sequences may reflect the stress-directed adaptive variations due to phenotype evolution that results from increases in mutation rates of genes responding to a particular stress[54]. Since there are three key divergences of modern humans, between Europeans and Asians, between Africans and non-Africans, and between the Khoe–San of southern Africa and other modern humans[55], the insertion regions of the common sequences could be the hotspots that promote evolution not only for Chinese but also for Eastern Asians.

In summary, we first constructed a two-component pan-genome that could represent the Han Chinese genome sequences missed from the current reference. Our growth analysis suggested that the size of common sequences is not bounded by an upper limit with an increasing population when using a constant occurrence threshold. Instead, we introduced the concept of occurrence frequency (OF), which suggested the possibility of constructing bounded-size pan-genomes by dynamically adjusting the threshold based on the sampled population. We realized the limitation of our conclusion using only 486 individuals. With more individuals studied, more individual-specific sequences defined in this study would turn out to be common sequences. But meanwhile, more individual-specific sequences will also be found. Our current study overthrew the pan-genome size estimation we made in 2010[19] (estimated a complete human pan-genome contains ~19–40 Mbp of novels sequence missing from the reference genome NCBI Build 36.3), and provided a new perspective of estimating pan-genome size. However, its accuracy remains to be verified by more whole-genome sequenced Chinese individuals in the future. The distinct characteristics and application efficiency of the CPG common sequences offer a high possibility of the 46.646 Mbp sequences as new "decoys" and are likely supplementary to the current human reference genome. We suggested a possible connection of the common sequences with population-specific evolution, yet the biological significance of most CPG sequences remains unknown and needs to be interpreted. With an accumulating amount of deep sequenced data becoming available, current pan-genomic studies will be extended to be more comprehensive population-scale and to explore the biological meaning of the detected genetic variations.

## Methods

**Assembly and filtering of novel contigs**. We used BWA (version 0.7.12) to align the raw reads (paired-end, 150 bp) of the 486 individuals to GRCh38.p13 and obtained all unaligned reads by SAMtools (version 0.1.19)[56]. Read pairs with at least one unaligned read were classified as unaligned pairs (both reads were unaligned) or semi-aligned pairs (just one read was unaligned). All unaligned reads were assembled into contigs using MEGAHIT (version 1.1.1)[57].

To confirm the uniqueness of the assembled contigs, we aligned the contigs to GRCh38.p13 containing patches and alternative locations. All contigs mapped with ≥90% identity and ≥80% coverage were discarded. We aligned the remaining contigs to non-cordate genomes by BLASTN (version 2.9.0) and removed the sequences aligned with >90% identity and >95% coverage to ensure that no contaminant sequences remained in the novel contigs (Supplementary Note 1, Supplementary Data 4). Contigs aligned to EmVec and UniVec sequences with ≥80% identity and ≥50% coverage were also discarded.

**Positioning of contigs in GRCh38**. All unaligned reads in semi-aligned pairs were mapped to the remaining contigs by Bowtie2[58]. For every contig, we needed to know whether each of its two ends could be positioned in GRCh38. For each contig end, we considered all reads mapped to within 150 bp of that end. Then, we collected the linked mates of those reads and checked their mapped positions. If there were more than 2 mapped linked mates and at least 95% of them were mapped to the same 2000-bp genomic region, the contig end would be classified as "exactly placed" (or "placed" for short).

The next step was to determine the exact position of the placed end in GRCh38. We considered the aforementioned 2000-bp genomic region in which the linked mates were mapped (500 bp would be attached to the start and end of this genomic region respectively). We aligned the terminal 100 bp of the placed end to the region using NUCmer (version 3.2.3)[59] (with options–minmatch 15–breaklen 1). If at least one alignment started within 5 bp of the placed end and all alignments of the contig were consistent with one another, the exact position of the placed was then derived from the alignment result.

We classified contigs based on their placement status. If both ends of a contig were placed on the same chromosome with the same orientation, we called the contig both-end-placed (BEP). If only one end of the contig was placed, we called it single-end-placed (SEP). We further divided SEP contigs into left-end-placed (LEP) and right-end-placed (REP) contigs depending on which end of the contig was placed when aligned to the positive strand of GRCh38. If neither end of the contig was placed, we called it an unplaced contig.

We defined the placement location of an LEP as the leftmost base-pair of the leftmost alignment and the placement location of an REP as the rightmost base-pair of the rightmost alignment. The insertion position of an LEP was defined as the rightmost base-pair of the leftmost alignment and the insertion position of an REP as the leftmost base-pair of the rightmost alignment. The placement location and insertion position of each end of a BEP contig are defined similarly as an LEP or REP. Note that the two insertion points of a BEP contig may coincide.

**Placed contig clustering**. Based on placement positions and sequence similarity, we grouped the placed contigs of 486 individuals into clusters as follows:

*Both-end-placed contigs*. For BEP contigs placed within 20 bp of each other, we used BEDtools (version 2.25.0)[60] to cluster them together and selected the longest one as the representative of the cluster. Next, we aligned all contigs in the cluster to the representative by NUCmer (using default parameters). Contigs that could not be aligned to its representative were removed from the cluster. To make the clusters more complete, SEP contigs and unplaced contigs were aligned to contigs in BEP clusters by BLASTN. These contigs were added to the BEP cluster if they were fully contained with >99% identity, and covered >80% of the aligned BEP. If the coverage of the BEP contig was under 80%, the SEP or unplaced contig was added to the BEP cluster only if it met two conditions: (1) the SEP or the unplaced contig had at least 5 mapped linked mates and (2) at least 25% of its mapped linked mates were within 2000 bp of the BEP cluster's placement location.

*Single-end-placed contigs*. We processed LEP contigs and REP contigs separately. As in the case for BEP contigs, we clustered LEP contigs placed within 20 bp of each other and selected the longest one as the cluster representative. LEP clusters within 20 bp of the placement location of any BEP cluster were discarded. For each discarded LEP cluster, its contigs were aligned to the corresponding BEP cluster's representative using BLASTN. Contigs aligned with >90% identity and >80% coverage were added to the BEP cluster. Next, we removed from LEP clusters any contig which could not be aligned to its representative using NUCmer. To make the clusters more complete, unplaced contigs (which were not added to BEP clusters) were aligned to representatives of LEP clusters using BLASTN. An unplaced contig aligned with 100% coverage, 99% identity and covering 80% of the representative would be added to the LEP cluster. If the coverage of the representative was under 80%, the unplaced contig was added to the LEP cluster only if it met two conditions: 1) the unplaced contig had at least 4 mapped linked mates and 2) at least 25% of the mapped linked mates were within 2000 bp of the LEP

cluster's placement location. These steps were repeated for REP contigs to obtain REP clusters.

Thereafter, we detected whether LEP contigs overlapped with REP contigs to form single longer contigs. If an LEP contig and an REP contig were within 100 bp in the same orientation, we aligned the two contigs with each other by NUCmer. Based on the alignment result, the two contigs were merged to generate a new insertion by PopIns[61] (Supplementary Methods).

To guarantee non-redundancy of our clusters, we aligned all representatives to each other using BLASTN. If two representatives could align to one another with >98% identity and >95% coverage and their placement locations were less than 2000 bp apart, their corresponding clusters would be merged, with the longest contig in the merged cluster becoming the new representative (BEP contigs taking priority over SEP contigs).

**Unplaced contig clustering.** For the remaining unplaced contigs which were not added to clusters in the above steps, we only kept their non-repetitive portions, since highly repetitive sequences without insertion points were likely due to contamination. We built a workflow to mask the repetitive portions using Dustmasker[31], Trf (version 4.0.9)[62], and RepeatMasker (version 4.0.7)[63]. For unplaced contigs with a continuous repeat percentage greater than 50%, we removed the masked portions and retained the leftover sequences longer than 200 bp. Lastly, we utilized cd-hit-est version 4.8.1 (with options "-c 0.9 -n 8") to cluster the resulting (non-repetitive) unplaced contigs and obtain their representatives.

**Calculation of Chinese pan-genome size.** We performed an experiment to study the growth of the Chinese pan-genome size with respect to population size. In the experiment, we estimated the pan-genome size for population size $p$ by sampling $p$ individuals out of 486. We then identified all individual-specific sequences and common sequences among these $p$ individuals and calculated their total sizes, denoted by $I_p$ and $C_p$, respectively. Note that common sequences are defined as those appearing as least twice in the $p$ individuals, thus a common sequence for 486 individuals may not be a common sequence for $p$ individuals. The values of $I_p$ and $C_p$ were averaged over 1000 repetitions of sampling. We performed the experiment for $p$ from 1 up to 486 and the results are depicted in Fig. 3a.

To study how common sequences were affected by the occurrence threshold, the above experiment was repeated for different occurrence thresholds from 5 up to 200. The experiment results for CPG and APG are shown in Figs. 3b and 3c respectively.

**Pathogenicity of insertion regions of novel sequences.** We used the C-scores of Single Nucleotide Variants (SNV) provided by CADD (version 1.6, link: https://krishna.gs.washington.edu/download/CADD/v1.6/GRCh38/whole_genome_SNVs.tsv.gz) to compute the pathogenicity score of each insertion point. For any genome position, CADD provides the score of three candidate SNVs (e.g., with reference allele A, CADD will provide the score of three SNVs including "A → T", "A → C" and "A → G"). The pathogenicity score of the insertion point of a SEP contig was calculated as the average of the six scores of the adjacent 2 bp. The pathogenicity score of a BEP contig was calculated as an average of the scores of all the base pairs between the two insertion points. The pathogenicity score of each contig is listed in Supplementary Data 5. Only those contigs inserted to the primary sequences (i.e., chromosomes 1 to 22, X and Y) of GRCh38 were counted.

**Statistics and reproducibility.** All of our statistical analyses were carried out on biologically independent samples ($n = 486$). The statistical assumptions of the growth model of common sequence are detailed in Supplementary Note 2. Data presented on logarithmic scale were log-transformed prior to analysis.

**Reporting summary.** Further information on research design is available in the Nature Research Reporting Summary linked to this article.

## Code availability

Commands and parameters are included in Supplementary Note 10. Custom scripts used are available at https://github.com/HKU-BAL/CPG, or Zenodo with https://doi.org/10.5281/zenodo.5155074[64].

## Data availability

The novel sequences of CPG (1) have been deposited to the Genome Sequence Archive for Human (http://bigd.big.ac.cn/gsa-human/) at the BIG Data Center, Beijing Institute of Genomics, Chinese Academy of Sciences, under the accession number PRJCA003657, following the regulations of the Human Genetic Resources Administration of China (HGRAC); (2) are also available via email request. The novel sequences of 910 Africans in APG were downloaded from NCBI with accession number PDBU01000000. The assembly of HX1, KOREF1.0, and NA18507 were downloaded from NCBI with accession number GCA_001708065.2, GCA_001712695.1, and GCA_000005465.1, respectively. The raw reads of the 90 Han Chinese from BGI were downloaded from EBI with accession number PRJEB11005. Supplementary Fig. 6 and Supplementary Table 4 are

cited in Supplementary Note 4. Supplementary Data 7, Supplementary Data 8, and Supplementary Data 9 are cited in Supplementary Note 5. Supplementary Fig. 7 is cited in Supplementary Note 6. Supplementary Table 5 is cited in Supplementary Note 7. Supplementary Fig. 8 is cited in Supplementary Note 8. Supplementary Fig. 9 is cited in Supplementary Note 9.

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

## Acknowledgements

We thank Steven Salzberg for his insights and comments on the manuscript. Ruib. L. was supported by the ECS (grant number 27204518), and TRS (T21-705/20-N and T12-703/19-R) of the Hong Kong SAR government, and by the URC fund at the University of Hong Kong. Ruiq. L. and S.T. were supported by the Science and Technology Personnel Training Project of Capital of China (grant number Z191100006119007). B.Y. was supported by the General Research Fund (grant number 17117918) of the Hong Kong SAR government.

## Author contributions

S.T. and Ruiq. L. collected data. Q.L., B.Y. and Ruib. L. performed data processing and analysis. C.L. and Ruib. L. developed the modelling theory. T.L., Ruiq. L. and Ruib. L. conceived and advised the project. All authors wrote and approved the final manuscript.

## Competing interests

The authors declare no competing interests.
