## [Peer Review File · Communications Biology]

Reviewers' comments:

Reviewer #1 (Remarks to the Author):

This study used 486 deep-sequenced Han Chinese genomes to build a Chinese pan-genome (CPG). Novel contigs absent from the current human reference were assembled. The authors characterized these novel sequences and their insertion points in GRCh38. They reported improvements in alignment performance for previously unaligned reads using CPG. They also suggested a growth model based on occurrence frequency to estimate the growth of common sequences with sample size.

This manuscript is clearly-written and the methods were appropriate and described in details. The concept of the growth model is novel and the CPG is based on the largest sample size thus far and will be helpful to others who wants to improve mapping rate of their sequencing reads.

Readers will be benefited further if the authors consider making the following changes/clarification in the text/figures:

1. Define the concept of pan-genome
2. Line 110: Fig. 2A was referenced before Figure 1
3. Line 123: I did not get the logic behind the sentence "indicating the origin of the 486 individuals to be from a normal Han population"
4. Line 129: It is not intuitive why "unplaced contigs can be aligned closely to placed one" until I read the methodology
5. Line 206: It is not clear to me the source of mutation rate. Please include in methods if it is not already there.
6. Line 211 : I assume you mean enriched with regulatory regions
7. Line 223-225: clarify your logic behind "due to deletion" and "newly generated". Consider removing the arguments if not supported.
8. Line 229-230: It took me a while to understand how Figure 6A supports the statement. May be indicate the percentage in the text directly.
9. Line 442: Two mapped linked mates seem like a small number. What is the distribution of the number of mapped linked mates for each placed contig?
10. Line 240: What is the corresponding number of RefSeq sequences for these 24,300 hits?

11. Line 250: Define the databases hsat2,3 and alphoid
12. Line 288: Indicate the percentage as in line 280. Consider including the percentages (like those reported in line 280) in Table 2.
13. Line 316: Define HUPAN or consider using "HUPAN analysis pipeline"
14. Line 334: Do you mean excessively high or low?
15. Line 336: Sentence needs editing
16. Line 389: Be specific about the size estimation in 2010 that got overthrown
17. Fig 3A: Color not clear in legend. Consider using a bigger dot or a line. Y-axis indicated common sequences but I assume it means either individual-specific or common sequence.

Reviewer #2 (Remarks to the Author):

1. Brief summary of the manuscript

This paper presented the construction of a pan-genome from 486 deep sequenced Han Chinese genomes. A total of 276 Mbp novel sequences were identified with 46.646 Mbp common sequences shared by at least two individuals. The authors also found that the novel common sequences came from genomics regions with high mutation rates and low pathogenicity.

2. Overall impression of the work

This is a large size population specific genome sequencing data analysis study. Compared to the existing results, the overall size of novel sequences was multiple times larger, which required a reasonable explanation.

3. Specific comments, with recommendations for addressing each comment

1) In line 75-77, "These studies show that the sizes of the pan-genomes in humans are variable and therefore ill-defined and the studies do not indicate whether pan-genomes grow indefinitely as more individual inputs are sequenced", what were the exact reasons for the variable sizes of the human pan-genomes? Was it because they used different analysis strategies? Or different parameter settings of the same pipeline? In another words, are these variations of pan-genome sizes real or they are just results of different analysis pipelines? Comparing existing methods would help readers understand the underline problem.

2) By using "common sequences" as for the counting of total size of non-reference sequences, authors showed that the size of novel sequences is linear to the number of individuals, which was conflict with the next sentence that the total size of "common sequences" above a fixed percentage would remain constant.

3) Line 231-236, "CPG aligned to both APG and HCPG with a low mapping rate of less than 10%", these

mapping rates are very low, especially the mapping rate to HCPG. Any explanation?

4) Line237, "This result indicates the specificity of our pan-genome among Chinese and the genetic differences between the Chinese and African genomes." An alternative explanation of the small percentage of overlapped regions between the two Chinese pan-genomes was that the CPG size was way overestimated.

**Responses to
Comments of reviewers**

Building a Chinese pan-genome of 486 individuals

Qihui Li, Shilin Tian, Bin Yan, Chi Man Liu, Tak-Wah Lam, Ruiqiang Li, Ruibang Luo

COMMSBIO-20-3594-T

We thank for the constructive comments from the two reviewers. Using both the next-generation sequencing short-reads and single-molecule sequencing long-reads of three standard genomes provided by the National Institute of Standards and Technology (NIST) in the Genome In A Bottle (GIAB) project, we have done intensive genome assembly and analysis in the last three months and verified that, our novel sequence discovery pipeline has the best performance regarding the novel sequence completeness compared to previous pipelines. Below please find our point-to-point responses.

Reviewer: 1

This study used 486 deep-sequenced Han Chinese genomes to build a Chinese pan-genome (CPG). Novel contigs absent from the current human reference were assembled. The authors characterized these novel sequences and their insertion points in GRCh38. They reported improvements in alignment performance for previously unaligned reads using CPG. They also suggested a growth model based on occurrence frequency to estimate the growth of common sequences with sample size.

This manuscript is clearly-written and the methods were appropriate and described in details. The concept of the growth model is novel and the CPG is based on the largest sample size thus far and will be helpful to others who wants to improve mapping rate of their sequencing reads.

Readers will be benefited further if the authors consider making the following changes/clarification in the text/figures:

Comment 1. Define the concept of pan-genome

Reply to comment 1: Added in Line 59-60. "Pan-genome is a collection of all DNA sequences that occur in a species. Ideally, pan-genomic analysis better captures unexplored or missed variants to improve the decoding of the genetic basis of human diseases ¹."

Comment 2. Line 110: Fig. 2A was referenced before Figure 1

Reply to comment 2: Fixed. We accordingly changed the order of Fig1 and Fig2 in figure legend.

Comment 3. Line 123: I did not get the logic behind the sentence "indicating the origin of the 486 individuals to be from a normal Han population"

Reply to comment 3: We removed this unnecessary indication.

Comment 4. Line 129: It is not intuitive why "unplaced contigs can be aligned closely to placed one" until I read the methodology

Reply to comment 4: Added in place. “More definitions and details of placed and unplaced contigs are in Methods.”

Comment 5. Line 206: *It is not clear to me the source of mutation rate. Please include in methods if it is not already there.*

Reply to comment 5: We added a reference to Supplementary Note 3 – “Mutation rate near insertion points” in place.

Comment 6. Line 211 : *I assume you mean enriched with regulatory regions*

Reply to comment 6: Yes and fixed.

Comment 7. Line 223-225: *clarify your logic behind "due to deletion" and "newly generated". Consider removing the arguments if not supported.*

Reply to comment 7: The two unnecessary arguments are removed.

Comment 8. Line 229-230: *It took me a while to understand how Figure 6A supports the statement. May be indicate the percentage in the text directly.*

Reply to comment 8: Percentages added in place. “Based on a minimum requirement of 90% identity, the percentages of alignments to Chinese (23.7%) and Korean (24.5%) were much higher than to African (16.3%) (Fig. 6A).”

Comment 9. Line 442: *Two mapped linked mates seem like a small number. What is the distribution of the number of mapped linked mates for each placed contig?*

Reply to comment 9: The distribution is shown in the following two figures. The vast majority of placed contigs are with ≥ 5 mapped linked mates. Lowering the requirement to >2 is an empirical decision to further increase sensitivity.

Comment 10. Line 240: What is the corresponding number of RefSeq sequences for these 24,300 hits?

Reply to comment 10: The number is 6,138. The number is added to the sentence.

Comment 11. Line 250: Define the databases *hsat2,3* and *alphoid*

Reply to comment 11: Defined in place. “dna-brnn was used on APG and identified 85.1% of its sequences to be centromeric (82.4% Human Satellites 2 and 3 (*hsat2,3*); 2.7% alphoid satellite DNA (*alphoid*)).”

Comment 12. Line 288: Indicate the percentage as in line 280. Consider including the percentages (like those reported in line 280) in Table 2.

Reply to comment 12: Percentage added. “We further increased the sequencing depth by five times from 60-fold to 300-fold for HG005 and notably, the improvement persisted (false-positive variants reduced by 5.19%), ...” Percentages are also added to Table 2.

Comment 13. Line 316: Define HUPAN or consider using “HUPAN analysis pipeline”

Reply to comment 13: Defined HUPAN in place. “(HUMAN Pan-genome ANALYSIS system)”

Comment 14. Line 334: Do you mean excessively high or low?

Reply to comment 14: Fixed. “HUPAN relied on PCR-based sequencing data - the DNA sequences that contain either excessively high or excessively low GC may be partially missed from the result of HUPAN analysis.”

Comment 15. *Line 336: Sentence needs editing*

Reply to comment 15: Edited. "HUPAN was using SOAPdenovo2⁴⁶, which was developed by the corresponding author of this study. SOAPdenovo2 is an assembler that maximizes sequence contiguity in trade of sequence completeness, especially for those repetitive sequences that are more commonly observed in novel sequences."

Comment 16. *Line 389: Be specific about the size estimation in 2010 that got overthrown*

Reply to comment 16: Edited. "Our current study overthrew the pan-genome size estimation we made in 2010 (estimated a complete human pan-genome contains ~19-40 Mbp of novel sequence missing from the reference genome NCBI Build 36.3), and ..."

Comment 17. *Fig 3A: Color not clear in legend. Consider using a bigger dot or a line. Y-axis indicated common sequences but I assume it means either individual-specific or common sequence.*

Reply to comment 17: Edited in the Fig.3 legend.

Reviewer: 2

1. *Brief summary of the manuscript*

This paper presented the construction of a pan-genome from 486 deep sequenced Han Chinese genomes. A total of 276 Mbp novel sequences were identified with 46.646 Mbp common sequences shared by at least two individuals. The authors also found that the novel common sequences came from genomics regions with high mutation rates and low pathogenicity.

Comment 18. 2. *Overall impression of the work*

This is a large size population specific genome sequencing data analysis study. Compared to the existing results, the overall size of novel sequences was multiple times larger, which required a reasonable explanation.

3. *Specific comments, with recommendations for addressing each comment*

Comment 19. 1) *In line 75-77, "These studies show that the sizes of the pan-genomes in humans are variable and therefore ill-defined and the studies do not indicate whether pan-genomes grow indefinitely as more individual inputs are sequenced", what were the exact reasons for the variable sizes of the human pan-genomes? Was it because they used different analysis strategies? Or different parameter settings of the same pipeline? In another words, are these variations of pan-genome sizes real or they are just results of different analysis pipelines? Comparing existing methods would help readers understand the underline problem.*

Reply to comment 18 and 19: The reviewer is correct that the difference in novel sequence sizes of the human pan-genomes is because of different analysis pipelines. More specifically, it is the genome assembler used in a pipeline that determines the completeness of the discovered novel sequences. We designed the pipeline for this study to have superior novel sequence completeness than the previous pipelines. To prove it, we will rely on publicly available standard samples that have both

single-molecule sequencing long-reads and next-generation sequencing short reads. We know that long-read, if with sufficient coverage, provides better assembly sequence completeness than short-read (Shafin et al., 2020). If we use the novel sequences extracted from a long-read assembly as the baseline, we will be able to evaluate the completeness of the novel sequences extracted from the short-read assemblies of the same sample. In the following experiments, we assume those novel sequences found in the long-read assembly but unfound in a short-read assembly as authentic novel sequences missed by the selected short-read assembler. This assumption has a flaw that it overlooks the even-harder-to-be-assembled novel sequences that would be missed by using even long-read assembly, but it is sufficient for a comparative study of completeness of the short-read assemblies to support the validness of our study. We used three samples and their sequencing data, they are HG001 (Caucasian), HG002 (Ashkenazim) and HG005 (Han Chinese) provided by the National Institute of Standards and Technology (NIST) in the Genome in A Bottle (GIAB) project. The Oxford nanopore data of three samples were downloaded from *Link 1* and were all subsampled to 50x. The Illumina data were downloaded from *Link 2* and were also equally subsampled to 50x. For long-read assembly, we used the Shasta assembler (Shafin et al., 2020, *Link 3*). For short-read assembly, we compared two assemblers SOAPdenovo2 (Luo et al., 2012, *Link 4*), and MegaHit (Li et al. 2016, *Link 5*). SOAPdenovo2 was used in two previous novel sequence studies that have generated smaller novel sequence sizes, including “Building the sequence map of human pan-genome” (Li et al., 2010, analyzed two genomes, the corresponding author of this study was one of the co-first authors) and HUPAN (Duan et al. 2019, analyzed 275 genomes). MegaHit was the one used in this study to provide better assembly completeness. For Shasta and MegaHit, default parameters were used. For SOAPdenovo2, “-R” option was used, and the largest novel sequence size was chosen from the assembles of k-mer size 39, 49, 59 or 69. The same sequence filtration and extraction procedures described in the manuscript were applied to the assemblies of all three assemblers. We used a 48-core CPU machine for the assemblies and analyses, with each sample cost five days to two weeks. The results are summarized as follows.

Samples	Size (Mbp) of novel sequences found in			% of ONT Shasta found in	
	ONT Shasta	NGS SOAPdenovo2	NGS MegaHit	NGS SOAPdenovo2	NGS MegaHit
HG001	4.019	1.212	2.595	24.00%	53.46%
HG002	5.759	2.058	3.447	29.54%	47.42%
HG005	6.329	2.478	3.824	34.97%	50.45%

As shown in the table, of all three samples, Shasta has the highest sizes of novel sequences. SOAPdenovo2 has the lowest sizes, while MegaHit is about 1.6-2 times larger than SOAPdenovo2. The percentages of novel sequences in Shasta found also in Megahit (47.42-53.46%) are higher than those also found in SOAPdenovo2 (24.00-34.97%). These results concluded that MegaHit generates larger novel sequence sizes and gives better novel sequence completeness than SOAPdenovo2, thus ensuring the validness of the larger novel sequence sizes found in our study.

Noteworthy, as also discussed in detail in the manuscript, the corresponding author of this study is also the first author of SOAPdenovo2 and a co-first author of MegaHit. Thus, the author’s criticisms on his own tools can be trusted. SOAPdenovo2 was designed as an assembler that maximizes sequence contiguity in trade of sequence completeness, especially for those repetitive sequences that are more commonly observed in novel sequences. MegaHit, in contrast, was designed to maximize sequence completeness. Hence, by design, MegaHit outperforms SOAPdenovo2 in novel sequence discovery and ensures better completeness. We foresee that in the future, when single-molecule sequencing becomes cheaper than next-generation sequencing, more samples with single-molecule sequencing will be used for novel sequence discovery to provide even better novel sequence completeness. But for now, choosing the right short-read genome assembler should be our best chance for novel sequence discovery.

Link 1: https://s3-us-west-2.amazonaws.com/human-pangenomics/index.html?prefix=NHGRI_UCSC_panel/

Link 2: <ftp://ftp-trace.ncbi.nlm.nih.gov/giab/ftp/data/>

Link 3: <https://github.com/chanzuckerberg/shasta>

Link 4: <https://github.com/aquaskyline/SOAPdenovo2>

Link 5: <https://github.com/voutcn/megahit>

Comment 20. 2) By using “common sequences” as for the counting of total size of non-reference sequences, authors showed that the size of novel sequences is linear to the number of individuals, which was conflict with the next sentence that the total size of “common sequences” above a fixed percentage would remain constant.

Reply to comment 20: Novel sequences can be either individual-specific or common. A linear growth in novel sequences does not necessarily imply an increase in common sequences. To help to understand why the total size of common sequences would remain constant with a fixed percentage of occurrence within a population, i.e., is determined by the occurrence frequency (OF), we consider a large population of N individuals. Let s be a sequence and f_s be the number of individuals in the population carrying s . Suppose that we use an OF of p ($0 \leq p \leq 1$). Now, we uniformly sample n individuals from the population. The expected number of individuals in this sample carrying sequence s is nf_s/N . Hence, according to the definition of OF, s will be a common sequence in this sample if and only if $nf_s/N > np$, in other words $f_s/N > p$. Note that this condition does not depend on the sample size n . Therefore, the set of common sequences defined by a particular OF is independent of the sample size, given that the sampling from original population is unbiased.

On the other hand, we claim that the total size of common sequences is unbounded under constant occurrence threshold (absolute number rather than percentage) as the population size increases. To explain the intuition behind our claim, we use a simplified tree model for population growth. We model the generations of a population as levels of a tree. The population starts from a single person, represented by the root node of the tree, in level 0. The root's offspring is represented by its child nodes in level 1. The offspring of a level 1 node is represented by its child nodes in level 2 and so on. For ease of explanation, we assume the tree to be a full binary tree, i.e. each non-leaf node has exactly two children. Suppose that there are m levels (0, 1, ..., $m-1$). The number of nodes in level k is then 2^k and the total number of nodes in the tree is $N=2^{m-1}$.

We also use a simplified model for sequence inheritance. We assume that each node carries a fixed number of sequences, which are then inherited by its children with a certain probability. In addition to sequences inherited from its parent, each node also has its novel sequences not found in its ancestors. We say that a sequence “originates” from a node if it is that node's novel sequence. We further assume that all novel sequences are unique, i.e. no two nodes will share the same novel sequence by chance. Under this model, a sequence originating from some node x can only be inherited by nodes within the subtree rooted as x . As an example, a sequence with 100% occurrence in the population must originate from the root (level 0); a sequence with 50% occurrence originates from either of the root's children (level 1); a sequence with 25% occurrence originates from one of the level 2 nodes and so on. On the other hand, a sequence originating from a leaf node (level $m-1$) has occurrence 1 (not 1%); a sequence originating from a leaf's parent node (level $m-2$) has occurrence 2 and so on.

Now let us examine common sequences defined using occurrence threshold (absolute number). For example, using an occurrence threshold of 4, the common sequences will be those originating from levels 0 to $m-3$. The total number of nodes in these levels is $2^{m-2}-1$, which is roughly $N/4$. Now consider doubling the population by introducing a new level m with 2^m people. With an occurrence threshold of 4, the common sequences will be those originating from levels 0 to $m-2$. The total number of nodes in these levels is $2^{m-1}-1$, which is roughly $N/2$. Hence the number of common sequences also doubles given the same occurrence threshold.

Next, we examine common sequences defined using occurrence frequency (OF). For example, using an occurrence frequency of 12.5%, the common sequences will be those originating from levels 0 to

3, which contain only 15 nodes. After doubling the population, the common sequences for 12.5% OF will still be those from levels 0 to 3. Hence the number of common sequences remains unchanged given the same OF.

In summary, if we use a constant occurrence threshold, the common sequence size has a linear relationship with the population size, therefore it is unbounded as the population grows. However, if we use a constant OF, the common sequence size is invariant of the population size, which matches our observation in the “Result” section. These explanations are also included as note 2 in the supplementary materials.

Comment 21. 3) Line 231-236, “CPG aligned to both APG and HCPG with a low mapping rate of less than 10%”, these mapping rates are very low, especially the mapping rate to HCPG. Any explanation?

Reply to comment 21: The mapping of this part included all novel sequences (including both common sequences and individual-specific sequences). The mapping rate is low as expected because when the number of individuals gets higher, the size of individual-specific sequences becomes dominant, as we observed and shown in Fig 3A. Similar numbers were also reported in the APG (Sherman et al., 2019) and HCPG (Duan et al., 2019) studies.

Comment 22. 4) Line237, “This result indicates the specificity of our pan-genome among Chinese and the genetic differences between the Chinese and African genomes.” An alternative explanation of the small percentage of overlapped regions between the two Chinese pan-genomes was that the CPG size was way overestimated.

Reply to comment 22: Please see “Reply to comment 18 and 19”.

Reviewers' comments:

Reviewer #1 (Remarks to the Author):

The authors have addressed all my comments.

Reviewer #3 (Remarks to the Author):

Although the authors have input a lot of efforts to show the different assembly results from assemblers like ONT Shasta, SOAPdenovo2 and MegaHit, it can not explain the huge size of the pan-genome they constructed. HUPAN did not use any of these assemblers they tested. In addition, the size of novel sequences per individual reported in the HUPAN paper was about 5Mb fully unaligned sequence and 6Mb partially unaligned sequences, which was much larger than the sizes reported here using SOAPdenovo2 (between 1.2 Mb to 2.4Mb).

Other comments.

- 1) In line 315, "In HUPAN (a Han Chinese pan-genome built by the Human Pan-genome Analysis system)", should be better to use "Using the HUPAN analysis pipeline".
- 2) In line 333, "HUPAN relied on PCR-based sequencing data", this is not accurate. HUPAN did not require PCR-based sequencing data as inputs.
- 3) In line 335, "HUPAN was using SOAPdenovo2", this was incorrect. HUPAN used SGA instead of SOAPdenovo2.
- 4) For the low mapping rate (<10%) of CPG to both APG and HCPG, this was not reported in APG or HCPG. Instead, about half of the novel sequences of HCPG can be aligned to APG, and almost one third are exactly the same.

**Responses to
Comments of reviewers**

Building a Chinese pan-genome of 486 individuals

Qihui Li, Shilin Tian, Bin Yan, Chi Man Liu, Tak-Wah Lam, Ruiqiang Li, Ruibang Luo

COMMSBIO-20-3594A

Dear Editor,

We thank the reviewer for her/his additional comments that helped us improve the statements of HUPAN's contribution in our manuscript. Below please find our point-to-point responses.

Reviewer: 3

Comment 1.

Although the authors have input a lot of efforts to show the different assembly results from assemblers like ONT Shasta, SOAPdenovo2 and MegaHit, it can not explain the huge size of the pan-genome they constructed. HUPAN did not use any of these assemblers they tested. In addition, the size of novel sequences per individual reported in the HUPAN paper was about 5Mb fully unaligned sequence and 6Mb partially unaligned sequences, which was much larger than the sizes reported here using SOAPdenovo2 (between 1.2 Mb to 2.4Mb).

Reply to comment 1:

We thank the reviewer for her/his additional comments that helped us to better identify HUPAN's methods and historical contributions in our revision. We have amended our descriptions about HUPAN in our revision regarding this and the next three comments. In the revision, we added:

1. "A more recent report named HUPAN, which is considered as the first pan-genome of Chinese, ..."
2. "We expected the use of PCR free data and MEGAHIT will largely maximize the sequence completeness of this study, and we indeed observed a larger size of novel sequences. But we also realized that not all these additional novel sequences are correct, and they need to be further validated through long-read assemblies or even wet-lab experiments in the future."
3. "HUPAN has used 185 deep sequencing and 90 assembled Han Chinese genomes. While the 185 deep sequencing genomes were PCR free and assembled using SGA, the 90 assembled Han Chinese were sequenced much earlier in 2013 with a PCR-based protocol and assembled using SOAPdenovo2."

But we disagree with "*HUPAN did not use any of these assemblers they tested*". In HUPAN's abstract, it wrote, and I quote: "We applied it to 185 deep sequencing and 90 assembled Han Chinese genomes and detected 29.5 Mb novel genomic sequences". While the 185 deep sequencing genomes were assembled using SGA, the 90 assembled Han Chinese genomes were assembled using SOAPdenovo2 [1]. In HUPAN, the authors wrote in the section named "Pan-genome analysis of 90 Han Chinese genomes": "After removing redundant sequences and potential contaminations, there were 10.37 Mb fully unaligned sequences left. When we aligned these sequences to the 30.72 Mb fully unaligned sequences from 185 deep sequencing genomes ...". It said that the 185 deep sequencing genomes generated 30.72Mbp sequences, while the 90 Han Chinese genomes only generated 10.37Mbp, which is disproportionate, let alone the increase of common sequences would slow down with more genomes. This is actually another proof from HUPAN that the selection of genome assembler matters, in addition to the statement "... we selected SGA (String Graph Assembler) due to its high assembly ..." written in the "De novo assembly" section in the HUPAN paper.

From our perspective, the larger size of the pan-genome that we have found in our study can be explained in three ways: 1) the algorithmic foundations and plenty of publicly available benchmarks of

the assemblers we used, 2) *in silico* validation using new types of data because more true novel sequences could be assembled using the new data type without needing more efforts on genome assembler, and 3) PCR validation of the novel sequences. About point 1, the authors of HUPAN selected SGA to assemble the 185 genomes they have sequenced themselves. I agree SGA was a good choice over SOAPdenovo2. And the use of SOAPdenovo2 on the other 90 genomes was inevitable because only the assembled sequences of the 90 genomes were provided [1]. But compare to MegaHIT, both SGA and SOAPdenovo2 are suboptimal choices for genome completeness. There are a few genome assembler benchmarking papers, and here I refer to a paper titled “Practical evaluation of 11 de novo assemblers in metagenome assembly” [2] that have covered all mainstream short-read genome assemblers in 2018. In the paper, table 3 titled “Performance of assemblers with real metagenomic data” showed, in terms of completeness, the total length of MegaHIT, SGA, and SOAPdenovo2 are 93, 63, and 65 Mbp, respectively. This explains why using MegaHIT generates more sequences than the other two assemblers, and the validity of those additional sequences was validated using real data [2]. About point 2, we have explained in detail in our previous response letter, and we want to reiterate that the completeness of long-read assemblies has an overall advantage over short-read assemblies, which is verified by many papers including but not limited to Shasta and Canu. Although point 3 (PCR validation) is interesting, we pray for understanding that it is beyond the scale and affordability of this study.

At the end of this reply, we want to emphasize again the contributions and the historical correctness of the selection of genome assemblers of HUPAN. We have made amendments of the statements in our revision. And as the author of two mainstream genome assemblers, I want to say that the development of genome assemblers is not ending. We expect to see short-read genome assemblers providing even better completeness in the future, and that will have a chance in making the sequences size of this study look small in turn.

[1] Lan T, et al. Deep whole-genome sequencing of 90 Han Chinese genomes. *GigaScience* 6, gix067 (2017).

[2] Forouzan E, et al. Practical evaluation of 11 de novo assemblers in metagenome assembly. *Journal of Microbiological Methods* 151: 99–105 (2018)

Comment 2.

Other comments.

1) *In line 315 · “In HUPAN (a Han Chinese pan-genome built by the Human Pan-genome Analysis system)”, should be better to use “Using the HUPAN analysis pipeline”.*

Reply to comment 2:

Amended.

Comment 3.

2) *In line 333, “HUPAN relied on PCR-based sequencing data”, this is not accurate. HUPAN did not require PCR-based sequencing data as inputs.*

Reply to comment 3.

Amended to be more specific.

Comment 4.

3) *In line 335, “HUPAN was using SOAPdenovo2”, this was incorrect. HUPAN used SGA instead of SOAPdenovo2.*

Reply to comment 4.

Amended to be more specific.

Comment 5.

4) *For the low mapping rate (<10%) of CPG to both APG and HCPG, this was not reported in APG or HCPG. Instead, about half of the novel sequences of HCPG can be aligned to APG, and almost one third are exactly the same.*

Reply to comment 5.

The size of the novel sequences aligned to APG is actually similar between our study and HCPG. Our study has 13.51 Mbp aligned to APG. HCPG has 14.65 Mbp aligned to APG. This actually indicates that both our study and HCPG have captured the essential common novel sequences that the Chinese are shared with Africans. The lower percentages are just a consequence of a larger size of novel sequences, and the percentages are not comparable across studies because of different sizes for the denominator. For better clarity, we have added the absolute sizes in place in our revision.

REVIEWERS' COMMENTS:

Reviewer #3 (Remarks to the Author):

The authors have addressed most of my questions. The Han Chinese pan-genome they constructed has a total size of 276 Mbp novel sequence which is much bigger (almost 10 times bigger) than the previous reported one. In addition, the Han Chinese pan-genome size was estimated to be open or infinite, which has not been reported for any Eukaryote. These are extraordinary results.

Minor revisions

1. HUPAN was a human pan-genome construction and analysis system, not a pan-genome. Please revise the manuscript accordingly.